# Self-Emulsifying Drug Delivery Systems: An Alternative Approach to Improve Brain Bioavailability of Poorly Water-Soluble Drugs through Intranasal Administration

**DOI:** 10.3390/pharmaceutics14071487

**Published:** 2022-07-18

**Authors:** Sara Meirinho, Márcio Rodrigues, Adriana O. Santos, Amílcar Falcão, Gilberto Alves

**Affiliations:** 1CICS-UBI—Health Sciences Research Centre, University of Beira Interior, Av. Infante D. Henrique, 6200-506 Covilha, Portugal; sara.meirinho@ubi.pt (S.M.); marciorodrigues@fcsaude.ubi.pt (M.R.); asantos@fcsaude.ubi.pt (A.O.S.); 2Faculty of Health Sciences, University of Beira Interior, Av. Infante D. Henrique, 6200-506 Covilha, Portugal; 3CPIRN-UDI-IPG—Center for Potential and Innovation of Natural Resources, Research Unit for Inland Development, Polythecnic Institute of Guarda, 6300-559 Guarda, Portugal; 4CIBIT/ICNAS—Coimbra Institute for Biomedical Imaging and Translational Research/Institute of Nuclear Sciences Applied to Health, University of Coimbra, Pólo das Ciências da Saúde, Azinhaga de Santa Comba, 3000-548 Coimbra, Portugal; acfalcao@ff.uc.pt; 5Laboratory of Pharmacology, Faculty of Pharmacy, University of Coimbra, Pólo das Ciêmcias da Saúde, Azinhaga de Santa Comba, 3000-548 Coimbra, Portugal

**Keywords:** bioavailability, brain, intranasal, neurotherapeutics, self-emulsifying drug delivery systems

## Abstract

Efforts in discovering new and effective neurotherapeutics are made daily, although most fail to reach clinical trials. The main reason is their poor bioavailability, related to poor aqueous solubility, limited permeability through biological membranes, and the hepatic first-pass metabolism. Nevertheless, crossing the blood–brain barrier is the major drawback associated with brain drug delivery. To overcome it, intranasal administration has become more attractive, in some cases even surpassing the oral route. The unique anatomical features of the nasal cavity allow partial direct drug delivery to the brain, circumventing the blood–brain barrier. Systemic absorption through the nasal cavity also avoids the hepatic first-pass metabolism, increasing the systemic bioavailability of highly metabolized entities. Nevertheless, most neurotherapeutics present physicochemical characteristics that require them to be formulated in lipidic nanosystems as self-emulsifying drug delivery systems (SEDDS). These are isotropic mixtures of oils, surfactants, and co-surfactants that, after aqueous dilution, generate micro or nanoemulsions loading high concentrations of lipophilic drugs. SEDDS should overcome drug precipitation in absorption sites, increase their permeation through absorptive membranes, and enhance the stability of labile drugs against enzymatic activity. Thus, combining the advantages of SEDDS and those of the intranasal route for brain delivery, an increase in drugs’ brain targeting and bioavailability could be expected. This review deeply characterizes SEDDS as a lipidic nanosystem, gathering important information regarding the mechanisms associated with the intranasal delivery of drugs loaded in SEDDS. In the end, in vivo results after SEDDS intranasal or oral administration are discussed, globally revealing their efficacy in comparison with common solutions or suspensions.

## 1. Introduction

Over the last years, the prevalence of neurological disorders has been increasing. According to the 2020 World Health Organization (WHO) report, neurological disorders affect up to one billion people worldwide [1]. In fact, global statistics show that 50 million people suffer from epilepsy, 62 million from cerebrovascular diseases, 326 million from migraine, and 24 million from Alzheimer’s disease and other dementias. For that reason, neurological disorders are considered one of the biggest causes of disability and death worldwide [1,2]. Taking this into account, efforts in discovering and developing new and effective neuropharmaceuticals are made daily, even though most of the new entities fail to reach clinical trials [3].

In fact, drug delivery to the brain depicts a great challenge—drugs should pass intact through absorptive membranes, circumvent the hepatic first-pass effect and, finally, cross the complex blood–brain barrier (BBB) [3,4]. To overcome absorptive membranes and BBB, molecules need to be lipophilic, with low molecular weight (<400 Da), nonionizable at physiological pH, and not substrates of active efflux transporters [3,5]. To fit these characteristics, approximately 40–70% of the new chemical entities investigated to treat neurological disorders end up belonging to the biopharmaceutical classification system (BCS) classes II and IV [6,7]. However, while class II drugs have poor aqueous solubility but high permeability at therapeutic doses, class IV drugs are characterized by their poor solubility and permeability. This can limit dissolution, absorption rate, and extension, consequently restricting bioavailability and drugs’ onset of action [8,9]. So, to achieve plasmatic therapeutic concentrations, there is a need to administrate high doses, further resulting in drug waste and in a possible increase in adverse effects and drug–drug or food–drug interactions [10].

To overcome the aforementioned problems, alternative formulations have been developed, with lipidic nanosystems gaining more interest over the last years. The main goal of these systems is to keep lipophilic compounds in solution after contact with aqueous environments, such as those present in the gastrointestinal tract (GIT) or in nasal mucosa [8]. Self-emulsifying drug delivery systems (SEDDS) are a type of lipidic nanosystem well-recognized for their ability to incorporate lipophilic BCS class II and IV drugs [8,9]. The successful commercialization of Sandimmun Neoral^®^, (cyclosporin A) (Novartis Pharmaceuticals, USA), Norvir^®^ (ritonavir) (AbbVie Inc., North Chicago, IL, USA), and Fortovase^®^ (saquinavir) (Roche, Basel, Switzerland) has inspired more investigation regarding SEDDS technology [11]. So, in addition to these, other products based on liquid SEDDS are currently commercialized, either encapsulated in hard gelatin capsules (Gengraf^®^ (cyclosporine) (AbbVie Inc., North Chicago, IL, USA), Lipirex^®^ (atorvastatin) (Highnoon Laboratories, Lahore, Pakistan)) or in soft gelatin capsules (Agenerase^®^ (amprenavir) (Glaxo Group, United Kingdom), Depakene^®^ (valproic acid) (AbbVie Inc., North Chicago, IL, USA), Rocaltrol^®^ (calcitriol) (Validus Pharmaceuticals, Parsippany-Troy Hills, NJ, USA), Targretin^®^ (bexarotene) (Eisai Co., Tokyo, Japan), Vesanoid^®^ (tretinoin) (Roche, Basel, Switzerland), Accutane^®^ (isotretinoin) (Roche, Basel, Switzerland), and Aptivus^®^ (tipranavir) (Boehringer Ingelheim, Ingelheim am Rhein, Germany)) [9,12]. Actually, most drugs belonging to BCS class II and IV could be incorporated into SEDDS. In that way, an improvement in their aqueous solubility and, consequently, in their absorption could be reflected in a bioavailability increase, without the need for high dose administration [13].

Until now, the core of SEDDS investigation has been directed to the oral route [14]. However, other routes can be of great interest for SEDDS administration, particularly if they allow a better brain targeting of central nervous system (CNS)-active drugs. In this context, the intranasal (IN) administration of drugs incorporated in lipidic nanosystems, such as SEDDS, could be a clinically beneficial alternative to explore. The main reason relies on the fact that the nasal cavity is the only anatomical area that directly connects CNS with the exterior. Therefore, this administration route becomes very attractive in the treatment of neurological disorders, since drugs can be partially transported directly to the brain, circumventing the BBB [15,16]. Drugs administered through the nasal cavity can also reach the brain by blood circulation. This allows a systemic drug absorption with no gastrointestinal passage and no hepatic first-pass effect [15,17,18]. Since the aqueous volume of the nasal cavity is very low compared with that of GIT, the risk of drug precipitation loaded in a SEDDS is reduced, although relatively potent drugs are still required. If drugs are unstable in acid environments, their IN delivery incorporated in SEDDS can also overcome that problem, since the pH range of nasal mucosa is between 5–6.5 [10]. So, gathering the potential of the IN route for nose-to-brain delivery, together with the advantages of formulating neurotherapeutics in SEDDS, a higher brain bioavailability and an improvement in the therapeutic management of patients might be expected. By doing so, numerous chronic complications associated with diseases and lack of therapy adherence might be avoided.

In this review, we deeply describe the concept of SEDDS as lipidic nanocarriers, focusing on all the important stages of their development. At this step, we discuss studies that incorporated neurotherapeutic agents in SEDDS, focusing our attention on their physicochemical and in vitro characterization. Next, we describe the mechanisms of brain drug delivery after IN administration of drugs loaded in SEDDS. Finally, we address the in vivo parameters obtained after IN or oral administration of neuropharmaceuticals loaded in SEDDS, enabling us to understand the potential of SEDDS in better treating neurological disorders compared with common pharmaceutical forms, such as solutions, suspensions or solid dosage forms.

## 2. SEDDS Development for Delivery of Neurotherapeutics Agents to the Brain

According to their composition and characteristics after aqueous dispersion, oral lipid-based solutions are classified as Type I to III [19]. Type III solutions, typically regarded as SEDDS, are transparent or semitransparent isotropic mixtures of oils, surfactants and cosurfactants in which drugs are solubilized [7,12,13]. For that, SEDDS can also be considered as the anhydrous preconcentrates of micro or nanoemulsions [20]. Depending on whether the oil proportion is higher or lower, they can be divided into subtypes IIIA or IIIB. SEDDS belonging to the subtype IIIA usually have a higher oil proportion in their composition, spontaneously originating nanoemulsions with a droplet size between 100–250 nm after dispersion in an aqueous medium. In comparison, subtype IIIB SEDDS are usually composed of less than 20% oil, having a higher proportion (20–50%) of both hydrophilic surfactants and organic hydrophilic cosurfactants. After the dispersion of subtype IIIB SEDDS in an aqueous medium, a microemulsion is spontaneously formed, originating microemulsions with a droplet size of fewer than 100 nm [19,21]. So, SEDDS belonging to the subtypes IIIA and IIIB can be, respectively, referred as self-nanoemulsifying drug delivery systems (SNEDDS) or as self-microemulsifying drug delivery systems (SMEDDS). Even so, when analyzing works published in the scientific literature, great care must be taken in interpreting these designations, since authors often use them imprecisely.

In terms of composition, lipophilic components are the most important excipients for SEDDS generation [12]. Usually, the selected oil is the one that demonstrates a maximum drug solubilization capacity, since it will influence the formulation loading capacity and drug absorption. Even though natural oils (e.g., soybean oil, sunflower oil, or olive oil) are preferable to be used for SEDDS design, they proved to exhibit poor drug-loading and emulsification capacities [9]. Therefore, medium- and long-chain triglyceride oils, with different degrees of saturation, have been preferably used to enhance drug solubility [7,12]. Surfactants are also very important in SEDDS composition because they help in the stabilization of the formed emulsions by reducing the surface tension between oil and the aqueous phase [7,22]. Commonly, surfactants are classified based on their charge (ionic or nonionic) and their hydrophilic–lipophilic balance (HLB) value. If they present a HLB > 10, they are classified as hydrophilic surfactants. Otherwise, when having a HLB < 10, they are classified as lipophilic surfactants [9,12]. Ideally, to enable self-emulsification with droplets having a particle size of fewer than 200 nm, surfactants should be nonionic and with an HLB > 12. When compared with ionic surfactants, nonionic hydrophilic surfactants are less toxic, also presenting a higher stabilization capacity in environments with a wide range of pH and ionic strength [9,12]. As demonstrated in Figure 1, all the here-reviewed studies that developed SEDDS for brain drug delivery have used nonionic surfactants, with the majority presenting HLB values > 12. In SEDDS composition, cosurfactants also play an important role. By promoting the dissolution of drugs and hydrophilic surfactants in oils, consequently decreasing interfacial tension, cosurfactants improve the emulsification process when in contact with aqueous phases [22]. This process leads to higher stability and homogeneity of the formed emulsions after aqueous dispersion [9,12,20]. Cosurfactants can also modulate SEDDS self-emulsification time and the droplet size of the formed emulsion [20]. Even though surfactants concentrations are usually between 30–60%, it is important that not only surfactants but also oils and cosurfactants are listed as generally recognized as safe (GRAS), consequently decreasing the risk of toxicity [7]. Once more, a point worth being discussed is that in the literature, the authors’ classification of SEDDS excipients as surfactants or cosurfactants does not always correspond to the expected. A clear example is the Tween 80 classification as a cosurfactant made by different authors [23,24,25] that, in reality, is a hydrophilic surfactant with an HLB of 15. So, for a better harmonization of excipients classification, in Figure 1 and Table 1 we considered as surfactants the hydrophilic surfactants with an HLB > 10, and as cosurfactants the organic solvents and hydrophobic surfactants (HLB < 10) used in addition to oils (sometimes hydrophobic surfactants are used as the oil component).

Apart from the components presented so far, in the SEDDS that have a very small percentage of water in their constitution, other ingredients such antioxidants (e.g., ascorbic acid), viscosity enhancers (e.g., chitosan), taste/odor masking agents (e.g., sorbitol, orange oil), and modified drug release ingredients (e.g., cellulose-based polymers) can be incorporated [9,25,26]. Another factor to be considered during SEDDS design is the characteristics of the aqueous biological medium with which they will come into contact. The main reason is that digestive enzymes, pH, and ionic strength of the aqueous medium might determine the droplet size of the formed emulsions in GIT or the nasal cavity [9,27]. A more dramatic impact on emulsions features can occur if the drug solubilized in SEDDS has a pH-dependent solubility [20]. In fact, drugs characteristics such as pKa, molecular weight, lipophilicity, presence of ionizable groups, and chemical structures, as well as the quantity of a drug loaded in the formulation, can have a considerable impact on SEDDS performance [20].

Regardless of the administration route used, SEDDS have several advantages in the drug delivery to CNS. Actually, when compared with ready-to-use emulsions, SEDDS are likely to have higher stability and are easier to produce in large-scale conditions [8]. As SEDDS provide the entire dose as a solution, for lipophilic drugs with limited dissolution rates and poor absorption, that can be highly advantageous to improve their solubility [7,13]. Additionally, as described above, after self-emulsification, the formed small droplets can generate a large interfacial surface, promoting the partitioning of drug molecules from the oil phase to aqueous environments and cell membranes interface. Furthermore, by encapsulation into the oil droplets, labile active agents can be protected from chemical and enzymatic destruction, being more effectively distributed throughout the body until reaching the biophase [13,21]. At the same time, the occurrence of irritant responses due to prolonged exposure of the nasal or GIT mucosa to active moieties can be overcome, since drugs are shielded into very small oily droplets. Nevertheless, even though oils, surfactants and cosurfactants used to develop SEDDS must be listed as GRAS, it is of the utmost importance to address the safety of either drugs, excipients and formulation itself in GIT mucosa and, even more importantly, in the nasal epithelium [7]. Emulsifying agents presenting high HLB values (e.g., Tween 80, Kolliphor EL—Figure 1) can promote the opening of tight junctions in both nasal and intestinal membranes, promoting the penetration of drug molecules contained in the formulation [8,21]. Nan et al. [43] evaluated this effect by measuring transepithelial electrical resistance (TEER) of Caco-2 cells monolayers during permeability studies. After the experiment, a reduction in TEER values was recorded, which can be explained by SMEDDS affecting the paracellular route through the opening of tight junctions. However, 48 h after the permeability study, TEER values of Caco-2 cells recovered, revealing a restoration of cell integrity [43]. Some emulsifying agents also exhibit inhibitory effects on efflux transporters such as P-glycoprotein (P-gp), increasing the bioavailability of efflux pump substrates. SEDDS can also increase drug uptake to the systemic circulation after oral administration since hepatic first-pass metabolism is avoided due to the lymphatic targeting of SEDDS after intestinal absorption [12,13,21]. In the case of nasal administration, this is not applicable since the own anatomical features of nasal blood supply enable a bypass of hepatic first-pass effect after drugs’ systemic absorption [18].

Although SEDDS offers all the described benefits, some limitations must also be considered. Particularly after oral administration, some care must be taken regarding possible precipitation of the solubilized drugs at the time of aqueous self-emulsification. This is more probable to occur in GIT than in the nasal cavity, since the aqueous volume is much higher. Regarding liquid SEDDS (L-SEDDS) for oral administration, they should be first filled in hard or soft gelatin capsules. This can lead to some long-term incompatibilities between the components of formulations and capsule shells, causing formulation leakage and lipidic oxidation of drug molecules and SEDDS components [8,20]. Moreover, the transfer of volatile cosolvents into the hard gelatin capsule shells might cause the precipitation of lipophilic active moieties. Altogether, this will lead to handling, storage, and stability problems [8,20]. Still, the conversion of L-SEDDS into solid SEDDS (S-SEDDS), and the addition of antioxidants to the formulation, can mitigate these disadvantages [8,9]. In the case of SEDDS nasal administration, the disadvantages regarding the encapsulation of L-SEDDS are not applied since nasal preparations are usually formulated as portable nasal droplets, sprays or gels.

### 2.1. Preparation and Physicochemical Characterization

In terms of the preparation process (Figure 2), SEDDS are easily prepared by simply mixing all the components. Then, the drug is added and solubilized to obtain a clear liquid preparation. At this step, L-SEDDS can remain in liquid form or be transformed into S-SEDDS. All different transformation processes implicate the addition of a solid carrier—adsorption, spray drying, hot-melt extrusion or freeze-drying [7,12,20]. Regardless of the state for which SEDDS are intended, the characterization of different physicochemical properties is mandatory to ensure the quality of the final preparation.

In this scope, it is of the utmost importance to evaluate the droplet size, polydispersity index (PDI), and zeta potential. These parameters are usually measured by dynamic light scattering after SEDDS dilution in an aqueous medium, even though other techniques can also be used (e.g., coulter counting) [9,12,13]. Droplet size and distribution of the formed emulsions are two of the most important characteristics in SEDDS design, as they can have a great impact on permeation. In vivo performance is also affected by these properties since droplet size and PDI determine the rate and extent of drug release and absorption pathways. Actually, if droplet size is small and homogeneously distributed, providing a larger interfacial surface, enhanced absorption of a drug could be achieved [7,8,9]. Even though these properties can change with the amount of drug loaded in the formulation, the percentage of oil in SEDDS is critical for the emulsion’s droplet size and PDI. This is quite evident from the work of Chen et al. [4] that obtained nanoemulsified butylidenephthalide (Bdph) nasal formulations with droplet sizes between 34.99 nm (17.2% Bdph) and 3760.89 nm (50% Bdph). In this case, Bdph is not only used as a therapeutic agent but also as an oil phase, explaining the marked differences in droplet size depending on the different Bdph proportions used. Nonetheless, except the optimized Bdph nasal SNEDDS [4] and the optimized intravenous (IV) teniposide SMEDDS [34], most of the works reviewed here reported particle sizes of fewer than 200 nm (Table 1). Zeta potential is another critical parameter for studying the surface charge of the droplets formed after dispersion [7,8,9,12]. It also provides information regarding colloidal stability, since high values of zeta potential (±40 mV) are translated into repulsive electrostatic forces between the formed droplets, avoiding particle aggregation and phase separation [9]. The surface charge can also have a high impact on the diffusivity of SEDDS after administration. This is mostly due to the sialic acid and sulfonic residues of the intestinal and nasal cavity mucosa that, at physiological pH, are negatively charged. So, as other authors already discussed, if droplets are neutral or slightly negatively charged, avoidance of ionic interactions with mucous components is expected [13,22]. By observing Table 1, it is evident that most SEDDS revised herein are negatively charged. This might prevent the absorption of the formed droplets through the negatively charged endothelial membranes. For that, a positive charge might be advantageous to promote endocytosis of the emulsion particles. Furthermore, and particularly when SEDDS are administered by IN route, a positive charge can increase the formulation residence time, since it can promote mucoadhesion and, consequently, decrease the mucociliary clearance [26,52]. Still, even though not extensively discussed in the scientific literature, probably other mechanisms such as hydrogen bonding or Van der Waals interactions might interfere with mucous interaction, being able to even overcome the strength of ionic interactions. Another parameter with a high impact on residence time is the L-SEDDS viscosity. This is usually assessed using rotational cone–plate or spindle viscometers [8]. SEDDS residence time is more dependent on viscosity when they are intranasally rather than orally administered. In fact, there should be a balance in viscosity values, since higher values can increase residence time and, consequently, the amount of drug absorbed in the nasal cavity. However, if formulations are too thick, drug release from the emulsions is compromised and the regular functions of cilia are hindered [53]. Care must be taken during viscosity characterization since the viscosity of undiluted and emulsified SEDDS can vary. This is shown in the Srivastava et al. [29] study where the undiluted viscosity of α-pinene SNEDDS is approximately 330-fold higher than the viscosity values obtained after 1:50 and 1:100 dilutions (Table 1). The pH of SEDDS after dilution can also be evaluated, particularly if aimed to be intranasally administered. Considering that the pH range of nasal mucosa is between 5 and 6.5 [10], to avoid nasal irritation and local toxicity, after self-emulsification, the pH of the formed emulsions should be within this interval. Of the studies revised here that developed SEDDS for nose-to-brain targeting, only Meirinho et al. [28], Chen et al. [30], and Nagaraja et al. [5] investigated the final pH of, respectively, SMEDDS loaded with perampanel, the temperature and pH-responsive in situ gel containing SMEDDS loaded with huperzine A, and naringin in situ gelling SNEDDS, being all within acceptable values (Table 1). The morphology and percentage of transmittance can also be examined to explore the shape and the clearness of SEDDS after suitable dilutions [7,8]. If the transmittance of a SEDDS is close to the water’s, it indicates the formation of a monotropic system and a complete miscibility of all components with each other [25]. The emulsification time is another important parameter, especially investigated for oral SEDDS. A possible explanation is the marked difference between the total volume that the human stomach can hold compared with the total volume of the nasal cavity [16,54]. Thus, depending on where SEDDS are intended to be dispersed, differences in droplet sizes, PDI, and emulsification time can be obtained [41]. The quantity of oil in a SEDDS can also interfere with emulsification time. In fact, Miao et al. [41] demonstrated that formulations up to 30% *w*/*w* in oil content showed an emulsification time of fewer than 90 s. By increasing oil percentage over 40% to 60% *w*/*w*, the emulsification time was increased to more than 200 s [41]. To assess emulsification time, a USP II dissolution apparatus is filled with a volume of a specific fluid maintained at 37 °C under agitation. Then, the time required to visualize changes after SEDDS dilution (e.g., obtention of a clear dispersion after SMEDDS forming microemulsions) is defined as the emulsification time. Faster emulsification can lead to a quicker drug release and, consequently, to a rapid onset of action [8,9,12]. SEDDS stability can be studied over a short time under different stress conditions (e.g., high temperature or freeze–thawing cycles, exposure to light, and centrifugation). In addition, stability at expected storage conditions for a longer time period can be addressed. Changes in appearance, droplet size, PDI, pH, self-emulsification, dissolution profiles, and others are then evaluated. If there are no significant changes before and after exposure, SEDDS are considered stable for that period [8]. These studies are very important, particularly to determine the appropriate storage conditions. For instance, Poorani et al. [24] determined that levosulpiride SNEDDS maintained all its physical characteristics after centrifugation at 12,000 rpm, cooling at 8 °C, and freeze–thawing cycles. However, after exposure to 50 °C, SNEDDS did not remain stable, being necessary to store it at room temperature or below to ensure a good formulation quality [24]. Chemical stability assays at the physiologic conditions the drug will be exposed to are also very important, particularly when SEDDS are loaded with chemically labile drugs. A clear example of this is depicted in the Dai et al. [36] study. They formulated stiripentol, an acid-labile antiepileptic drug, in a SNEDDS to increase bioavailability after oral administration. After an 8 h exposure of either a methanolic stiripentol solution or a stiripentol SNEDDS to a 0.1 M HCl solution, only 37.48% of stiripentol remains in the methanolic solution. On the contrary, when formulated in a SNEDDS, stiripentol percentage was kept at 97% after 8 h of experimentation, demonstrating the ability of these lipidic nanosystems in protecting acid-labile drugs from degradation [36].

### 2.2. In Vitro and Ex Vivo Evaluation

As demonstrated in Figure 2, in vitro and ex vivo assays are essential during the SEDDS development process.

Cytotoxicity assessment in different cell lines is of the utmost importance. It allows to estimate the relative toxicity of a formulation, loaded or not with a drug, and also to establish an initial dose to test in the animal studies [55]. This enables a reduction in the number of animals used in in vivo assays and, consequently, compliance with the 3 Rs rule [55,56].

In vitro dissolution assays are also very important, particularly for SEDDS loaded with poorly water-soluble drugs. These assays are able to predict drug dissolution/release rate and extent in the nasal cavity or GIT after SEDDS administration [28]. Formulation composition and its characteristics (e.g., particle size and viscosity) can have a high impact on dissolution. As Poorani et al. [24] described, formulations with higher oil content and, consequently, larger globule size (28.5% *w*/*w* of castor oil with 312.3 nm) only release <80% of levosulpiride. On the contrary, the optimized SNEDDS (18% *w/w* of castor oil with 197.3 nm particle size) released 100% of the drugs after 60 min of the assay. A possible explanation is that a higher oil content yields emulsions with larger particle sizes and less surface area, affecting the diffusion process [24]. However, the key impact on dissolution profiles is the solubility of drugs in the different fluids that mimic the in vivo conditions, particularly for drugs having pH-dependent solubility [9]. An example of this dependence is given by the Abdelmonem et al. [23] study, describing a faster release of lamotrigine from SNEDDS in a 0.1 N HCl medium than when placed in a PBS medium. This difference between release profiles is explained by the pH-dependent solubility of lamotrigine [23]. Another example of this dependence is the case of ferulic acid release from a SMEDDS, which was approximately 80% in a pH 1.2 medium but 100% in a phosphate buffer pH 6.8 medium [31]. If for other reasons SEDDS are meant to be dissolved in another medium type rather than the physiological ones, the stability in those conditions must be assessed. A clear example is the teniposide-SMEDDS developed by He et al. [34] for IV administration. SMEDDS were dissolved in two different solutions commonly used to dilute commercial teniposide formulation before administration—0.9% NaCl and 5% glucose. They concluded that the teniposide-SMEDDS should only be diluted in a 5% glucose solution (0.1–0.4 mg/mL) and be used within 4 h since their stability is only guaranteed in those conditions [34]. Pharmacopeia dissolution apparatus is one of the most common approaches used to study drug dissolution and release profiles. For in vitro release assessment, dialysis bags are also used, particularly in a fluid/fluid barrier context [57]. Another suitable alternative to predict in vitro drug release from a formulation are the horizontal Ussing chambers using synthetic membranes as barriers [58]. An example that employed horizontal Ussing chambers is reported by Meirinho et al. [28], who studied perampanel release from a SMEDDS formulated for nose-to-brain delivery. The authors were able to predict perampanel release percentage and rate after 4 h of assay and compare it with perampanel release from a Transcutol HP solution used as a positive release control [28]. Even so, few studies describe the use of this approach for the prediction of drug dissolution and release.

In vitro and ex vivo permeation studies are other predictive assays usually performed during SEDDS development. Parallel artificial membrane permeability assay (PAMPA) and Caco-2 cells assays are the two models mostly used for in vitro permeability prediction [59,60]. Both are useful to estimate intestinal absorption using, respectively, artificial lipidic membranes and Caco-2 cells monolayers. Even though not currently applied for SEDDS permeability prediction in the nasal cavity, some of the literature already reported the use of both models to foresee the permeation of other lipid-based nanosystems after nasal administration [61,62]. After performing the first permeation screening tests using in vitro models, ex vivo permeation assays enables a better prediction of the absorptive ability of gastrointestinal and nasal mucosa membranes, reducing the use of living animal experimentation [55,63]. Ex vivo permeation studies can be carried out using buccal, sublingual, nasal, vaginal, intestinal or even another mucosa [64]. Franz diffusion cells are, at this time, the most common system used for these assays [65]. Yet, over the last years, Ussing chambers also started to be applied in ex vivo permeation studies. As Franz cells, if used in a vertical conformation, Ussing chambers are ideal to study fluid/fluid interfaces. For this, a portion of mucosa (e.g., mouse intestine) needs to be assembled between the chambers. Then, the diffusion of a drug loaded in SEDDS can be studied in the absorptive or efflux directions [58,65,66]. On the other hand, if an air/fluid interface (e.g., nasal mucosa) is studied, the horizontal conformation of Ussing chambers can be more adequate [58]. As permeation, the toxicity of SEDDS can also be evaluated either in vitro or in vivo/ex vivo. For an in vivo/ex vivo approach, different organs or mucosal membranes coming from treated animals are used. Then, after different periods of exposing membranes to the studied formulations, a histopathologic evaluation is performed and compared with negative toxicity controls. This is of the utmost importance, particularly for nasal mucosa. In fact, safety considerations, not only of the drug itself but also regarding the excipients within the formulation, can determine local and systemic side effects of a developed system [15].

## 3. Intranasal Administration as an Alternative to the Oral Route in Brain Drug Delivery

The oral route remains the most common choice for drug administration. It is estimated that oral preparations represent 90% of the human pharmaceutical formulation market [67]. It provides advantages such as ease and self-administration, suitability for long-term use, cost effectiveness, and ability for scale-up manufacturing [27,67]. However, after oral administration, drugs need to pass several GIT compartments with different characteristics in terms of pH, aqueous volume, and membrane structures [2]. In addition, drug solubilization within GIT is mandatory, since incomplete dissolution may lead to incomplete absorption, low bioavailability, and higher fluctuations in systemic concentration [9]. After oral administration, the hepatic first-pass effect and possible interactions with other coadministered drugs or food components should also be considered, since both can limit the oral bioavailability of drugs [12,67]. When the treatment of CNS disorders is the target, drugs should also pass an even more complex barrier—the BBB—to enter the brain [3]. Neurotherapeutics are mostly lipophilic, having poor water solubility and, consequently, erratic dissolution after oral administration. Even so, a percentage of the administered dose can reach the systemic circulation. However, due to other physicochemical characteristics such as high molecular weight, these molecules could be restricted in passing through the BBB, hardly achieving effective therapeutic concentrations in the brain [2,3]. Different types of lipidic nanosystems have been developed to overcome the disadvantages of oral administration of neurotherapeutics, SEDDS being one of those systems [13,22]. Even though SEDDS are more frequently investigated for oral delivery, an even more favorable alternative might be to associate SEDDS with IN administration [14]. In fact, over the last years, there has been a growing interest in the field of nose-to-brain drug delivery for the treatment of neurological disorders [15]. Fundamentally, that interest relies on the existence of a unique anatomical connection between the neuroepithelium of olfactory mucosa and the brain through the cribriform plate of the ethmoid bone, allowing the direct access of neurotherapeutics to the CNS [18]. In addition, the IN route also offers advantages such as ease of administration and little risk of injury, as a result being more patient-friendly compared with more invasive routes [16]. Better brain targeting being expected, a quick onset of therapeutic effects can also be achieved and, by administering lower doses, fewer systemic adverse effects might be triggered [15,16]. Due to the anatomical features of nasal blood supply, the absorption of drugs into the bloodstream is usually unavoidable. Still, IN administration allows circumventing the first pass hepatic metabolism of the oral route, which is a huge advantage for strongly metabolized drugs [18]. IN delivery is also associated with some disadvantages—restricted volume, short residence time caused by mucociliary clearance, and the presence of degrading enzymes and efflux transporters [16]. However, as previously discussed in Section 2, formulations such as SEDDS can have excipients in their composition that are able to circumvent these disadvantages [22]. Still, the existence of mucosal injuries must be evaluated, since it can possibly alter the local absorption of drugs after IN administration [16]. The possible local and systemic toxicity, not only of the drug itself but also of the different excipients within the formulation, should also be considered [15].

Although derived microemulsions (and some nanoemulsions) have been extensively tested [10], up to date, and as far as we know, there are only six works in the scientific literature that developed SEDDS for nose-to-brain delivery [4,5,28]. However, after balancing the pros and cons of oral and IN routes, the increasing interest in the IN route for treating CNS disorders is undeniable. For this, an increase in the development of new SEDDS for that purpose is an expected challenge.

### Delivery Mechanisms through the Intranasal Route of Drugs Loaded in SEDDS

The nasal cavity is mainly divided into vestibule, olfactory, and respiratory regions [15,18]. As represented in Figure 3, the primary pathways associated with the direct transport of drugs from the nose to the brain are located in the olfactory and respiratory regions, those being the only external anatomical areas directly linked to the CNS [15,16]. In the olfactory region, this connection is made through the cribriform plate, a perforated bone where nerves coming from the olfactory bulb connect it to the olfactory neuroepithelium [14]. The trigeminal nerves also link the nasal cavity to CNS. In this case, trigeminal nerves enter the brain primarily through medulla oblongata and pons, with endings at the olfactory epithelium level (Figure 3) [14,15]. Regarding the respiratory region, which comprises over 80–90% of the total nasal surface area, its epithelial composition is much more complex [14]. Here, trigeminal nerves are also present, with their axonal endings likewise linking the nasal cavity directly to CNS. Contrary to the olfactory region, where mucus turnover takes several days to occur, in the respiratory region the goblet cells are responsible for mucous secretion, with a turnover of around 15–20 min [16,22]. Consequently, the clearance of exogenous compounds is faster, making the olfactory region a more desirable region for drug formulation deposition rather than the respiratory surface. In fact, if instillation is performed deeper in this posterior region with an upward, tilted head position, a maximum exposure to the olfactory region might be assured, facilitating drug absorption directly the to brain [16,68]. Nevertheless, the complex vascular network and the ciliated and nonciliated respiratory epithelium, combined with microvilli, largely increase the surface area for drug absorption, particularly through the indirect systemic pathways [14,18]. In fact, it is in the nasal respiratory region that most systemic absorption is expected to occur [16,18,68]. In addition to the obvious avoidance of GIT passage, anatomical features of respiratory mucosa blood supply might also explain the lack of hepatic first-pass effect [18]. Consequently, after systemic absorption of drugs intranasally administered, a higher bioavailability might be expected.

The mucous layer overlying the olfactory and respiratory regions is critical for the formation of micro or nanoemulsions after SEDDS administration. When SEDDS are instilled in the nasal cavity, the aqueous mucous and the cilia flow movement provide the ideal conditions for self-emulsification (Figure 3). Then, when in contact with epithelial cells, the formed emulsion droplets can either release the encapsulated drugs or be transported through the same mechanisms as drugs are transported when in the nasal cavity [14]. This is mostly dependent on the physicochemical properties of SEDDS and the emulsions that are formed. The formulation viscosity is determinant for the amount of time that it can be in contact with mucosa before clearance [22]. Even more important is the size and PDI of the formed droplets after emulsification [14,52]. When administered into the nasal cavity, it is important to consider that emulsion droplets or drug molecules can be delivered by more than one pathway. The factors that influence the preference for one pathway rather than other depend on the formulation composition, the emulsions/released drugs’ physicochemical properties, and the site and methodology of SEDDS application [14,52,69]. As demonstrated in Figure 3, the olfactory nerve route allows drugs to be absorbed by three different mechanisms. The first involves the absorption of drugs intracellularly throughout the olfactory neurons, either by endocytosis or pinocytosis (1). In this case, the axons of olfactory nerves present a diameter that, in theory, enables the direct transport of nanoparticles (up to 700 nm) into the brain [14,69]. The second pathway is the extracellular transport, in which hydrophilic drugs cross nasal epithelia by paracellular spaces until reaching lamina propria (2). This transport comprises the tight junctions between endothelial cells or the spaces between the olfactory nerves and the sustentacular cells [14,15,18,69]. The third alternative is the transcellular/intracellular transport through the epithelial cells (3). In this case, small lipophilic molecules cross epithelial cells by passive diffusion or, in the case of larger moieties, by endocytosis/transcytosis. Then, drugs are disseminated interstitially or diffuse regionally to different brain regions through axonal or periaxonal transport. Considering the trigeminal nerve route, direct delivery of drugs to the CNS might occur either by intracellular (1) or paracellular/extracellular (2) pathways [14,17,18,69]. Here, the mechanisms and physicochemical properties of drugs entering by one pathway or by the other are the same as those described for intracellular and extracellular olfactory transport routes. However, the literature describes that in mice, molecules transported through the intracellular pathway take about 0.74 to 2.67 h to be transported across the olfactory nerve and between 17 to 56 h to diffuse along the trigeminal nerve [69]. Nasal-administered SEDDS or free drugs released from emulsions can also be absorbed systemically in the respiratory region. This is more relevant for drugs with a higher membrane permeability, as in the case of BCS class II, or emulsions with a suitable hydrophilic/hydrophobic balance [52]. As shown in Figure 3, emulsions or drugs can be absorbed from lamina propria either to the lymphatic system (4) or blood system (5), reaching systemic circulation. Then, if able to cross the BBB, drugs can also reach the brain, achieving their therapeutic effect [52,69].

## 4. Application of SEDDS in Brain Drug Delivery—Pre-Clinical Studies Perspective

It is known that in vitro studies are of great importance during the research and development of new medicinal products, which include SEDDS. Still, pharmacokinetics and toxicology studies in animals play a major role in predicting further bioavailability, efficacy, and safety in humans. Therefore, the present review was only focused on scientific works that carried out in vivo pre-clinical studies specifically to treat neurological disorders. Of those, the majority of them used rodents (rats or mice) as animal models, since they are known for being economical and convenient [9].

A small number of SEDDS-based products have already been successfully introduced into the pharmaceutical market, with none of them loading drugs for the treatment of neurological disorders. Until now, several in vivo assays using oral SEDDS have been carried out for neuropharmaceuticals brain targeting [4,6,23,24,25,29,30,31,32,33,34,35,36,37,38,39,40,41,42,43,44,45,46,47,48,49,50,51,70], but only a few in vivo pre-clinical studies have been reported for IN administration of neurotherapeutics loaded in SEDDS. In fact, only six studies were found that investigated the IN administration of neurotherapeutics loaded in SEDDS: clonazepam [45], diazepam [44], and perampanel [28] for epilepsy treatment; Bdph for glioblastoma treatment [4]; huperzine A [30] for Alzheimer’s disease; and naringin [5] as a neuroprotective for Alzheimer’s and Parkinson’s disease. All these studies are summarized in Table 2, with a brief description of the assessed limitations and the major in vivo outcomes after SEDDS administration. In Table 2, it can also be observed that the majority of the here-reviewed compounds belong to BCS class II or IV, similar to most chemical entities investigated to treat neurological disorders [6,7]. Only zolmitriptan [46], ginsenoside Rg1 [47], and chlorogenic acid [48] are reported to belong to BCS class III.

In general, all the works describing pharmacokinetic studies demonstrated an increase in brain concentrations and drug bioavailability using SEDDS. For comparison, conventional drug solutions, suspensions or commercial preparations were used. Only Kaur et al. [44] referred a lower plasma and brain diazepam bioavailability after nasal administration of SMEDDS, compared with the IV solution. Still, the same efficacy in treating status epilepticus could be achieved, since AUC_brain_/AUC_plasma_ ratios after nasal and IV administrations to rats were similar (Table 2) [44]. Abd El-Halim et al. [46] also reported lower zolmitriptan brain concentrations using a SNEDDS when compared with an oral solution. Still, physiological state and algesia, maintaining normal brain activity, were attained after oral administration of SNEDDS, reaching the same therapeutic efficacy with lower brain concentrations [46]. After IN administration, Meirinho et al. [28] and Nagaraja et al. [5] demonstrated, respectively, an increase in perampanel and naringin plasmatic and brain C_max_ and AUC_0-t_, compared with the corresponding suspensions. This increase in plasmatic and brain C_max_ and AUC_0-t_ values was particularly higher for naringin than for perampanel. Similarly, the relative bioavailability obtained for naringin was also higher than for perampanel (306.6% vs. 134.1%) (Table 2). Physicochemical and pharmacokinetic differences between both drugs and formulations should enable us to explain the differences, since they mainly determine drugs’ fate after administration. From the revised studies that developed IN SEDDS, Meirinho et al. [28] and Nagaraja et al. [5] were the only two that investigated the nasal tolerability of the developed SEDDS. For both cases, histopathologic results show no signs of toxicity in nasal mucosa (Table 2), which allowed us to consider both formulations as safe. Another approach to using SEDDS for epilepsy management, particularly status epilepticus treatment, was described by Vyas et al. [45]. Similar to what Meirinho et al. [28] reported for perampanel SMEDDS nasal administration, a direct brain targeting of clonazepam was demonstrated by the values of drug targeting efficiency (DTE) and direct transport percentage (DTP) higher than 100% and 0%, respectively (Table 2). Huperzine A loaded in a nasal SMEDDS temperature and pH-responsive gel allowed its sustained release behavior, obtaining a plasmatic and brain absolute bioavailability higher than 100%, [30]. Chen et al. [4] also investigated the use of the IN route for treating glioblastoma by administering SNEDDS loaded with Bdph. No pharmacokinetic evaluation was performed during this study, only demonstrating the efficacy of the SNEDDS in increasing the survival rate of rats with intracerebral malignant tumors using half the dose of that administered in a simple solution [4]. This demonstrates the advantages of using nasal SEDDS rather than a simple nasal solution, the oral route or the invasive IV route for brain delivery of drugs. By analyzing Table 2, it is noted that, despite the pharmacokinetic evaluation, some authors also performed behavioral studies to compare the efficacy and safety of drugs loaded in SEDDS administered with those of simple solutions, suspensions or commercial preparations of the same drugs [4,25,29,31,32,33,46,48,49,50]. For the case of glioblastoma treatment, efficacy was directly assessed after oral administration of chlorogenic acid SMEDDS [48] or the new prodrug 13a-(S)-3-pivaloylocyl-6,7-dimethoxyphenanthro(9,10-b)-indolizidine (CAT3) [49]. For this, tumor growth in rodent glioma models was evaluated either by weighting the tumor after euthanasia [48] or through bioluminescence signals [49]. Both studies concluded that loading the drugs into SEDDS leads to a larger inhibition of tumor growth compared with the oral suspension. In both cases, an increase in plasmatic C_max_ and AUC_0-t_ was attained with SEDDS oral administration [48,49] (Table 2). Liu et al. [31] evaluated the efficacy of ferulic acid SMEDDS in insomnia mice. An extension by 2-fold in sleep time was accomplished by SMEDDS’ oral administration compared with oral solution, which might not only be explained by the increase in both C_max_ and AUC_0-t_, but also by the increase in the elimination of the half-life (t_1/2_) of ferulic acid [31]. As chlorogenic acid, ferulic acid belongs to the phenolic acids group, being characterized by its instability, low absorption, and erratic bioavailability [71]. So, considering the ability of SEDDS in protecting labile drugs from degradation and increasing the permeability of low permeable compounds, the obtained in vivo results were expected to occur. Regarding other studies that performed behavioral assays, a clear example is demonstrated by Aswar et al. [50]. Here, the authors applied three different behavior studies—passive avoidance test, open field test, and ambulation counts—to evaluate the effect of curcumin-SMEDDS in treating Wistar rats with depression induced by bulbectomy. By this pharmacodynamic evaluation, the authors concluded that curcumin-SMEDDS had a better effect on improving memory, cognition, and locomotion, with a decrease in serum cortisol levels, compared with those receiving pure curcumin [50]. For treating Parkinson’s disease, Srivastava et al. [29] and Borkar et al. [33] also used behavioral studies to evaluate the therapeutic advantages of, respectively, α-pinene and dipalmitoyl apomorphine loaded in SEDDS. Compared with oral suspension, α-pinene-SNEDDS triggered a higher attenuation in tremulous jaw movements in the hypothermic effect and the inhibition of salivation and lacrimation, all induced by pilocarpine administration [29]. To study the effects of dipalmitoyl-apomorphine-loaded SEDDS, the 6-hydroxydopamine-lesioned rat model was applied. Compared with oral or subcutaneous apomorphine administration, the behavioral responses in lesioned rats were prolonged after oral dipalmitoyl apomorphine SEDDS administration, making it possible to conclude that SEDDS can provide a sustained drug release, leading to steady-state brain exposure [33]. 

As summarized in Table 2, most of the here-revised studies were focused on performing pharmacokinetic analysis after SEDDS administration. This is the case of the Miao et al. [41] study that developed a specific type of solid SNEDDS—sustained-release SNEDDS pellets—to increase the oral bioavailability of the atypical antipsychotic drug, ziprasidone. Since ziprasidone is highly susceptible to food effects, with consequent fluctuations in its plasmatic concentrations, pharmacokinetic studies in fed and fasted dogs were conducted. The SNEDDS pellets and commercial tablets of ziprasidone were administered to different animal groups for further comparison. Even though C_max_ was lower with SNEDDS than with commercial tablets, a significant increase in t_max_, accompanied by an increase in mean residence time (MRT), was accomplished after the oral administration of sustained-release SNEDDS pellets. Compared with the commercial capsules, an increase in AUC_0-__∞_ and in relative bioavailability (157.8% and 150.1% in fed and fasted state, respectively) was also obtained (Table 2). In addition, no food effect was observed after oral administration of SNEDDS, contrary to what occurred after capsule administration [41]. Aside from improved oral absorption, SEDDS have been reported to minimize the impact of food effects on drug dissolution through a unique combination between different formulation aspects and bioactive effects of some excipients [72]. To overcome some mutual limitations regarding antiepileptics oral administration, SEDDS were demonstrated to be a reliable alternative (Table 2). In fact, to overcome both low aqueous solubility, gastrointestinal instability, and GIT low permeability, lamotrigine, stiripentol, and carbamazepine were formulated as oral SNEDDS or SMEDDS [23,36,43]. For the three cases, both plasmatic C_max_ and AUC_0-t_ significantly increased compared with traditional oral formulations, which explains the obtained relative bioavailability higher than 100% (Table 2) [23,36,43]. Cannabidiol, the well-known therapeutic compound of the *Cannabis sativa* plant, was also formulated in a patented Swiss SEDDS technology—VESIsorb^®^ (SourceOne Global Partners, Chicago, IL, USA)—whose composition and characterization are scarcely available in the scientific literature [38]. Still, compared with an oil solution, there is a faster absorption together with an increase in plasmatic C_max_ and AUC after 8 h and 24 h of SEDDS administration (Table 2) [38].

## 5. Conclusions and Future Challenges

Considering the physicochemical characteristics of neurotherapeutics, SEDDS have been demonstrating a high potential in improving these drugs’ bioavailability, comparatively with traditional formulations such as tablets, solutions or suspensions. This is mostly due to the excipients that compose SEDDS, generating nanosized structures after dispersion in the aqueous medium. The large surface area of the formed droplets after aqueous dispersion enhances drug dissolution, permeability, and absorption through biological membranes, leading to higher bioavailability. This not only happens in GIT after oral administration but also if SEDDS are administered through the nasal route. Even though SEDDS have primarily been explored for bioavailability enhancement after oral administration, the application of SEDDS in nose-to-brain delivery is worth being explored more. This is mostly due to all the potential of the IN administration route in delivering drugs directly to the brain, circumventing the BBB. Thus, an even higher brain bioavailability of neurotherapeutics might be expected if these were intranasally administered loaded in SEDDS, compared with the oral route. However, as demonstrated in this review, little research has been developed regarding the field of SEDDS for IN administration. Hence, it is important to identify potential drugs to be loaded in SEDDS at higher concentrations, particularly those that are poorly water-soluble, to then be delivered by the IN route. Nevertheless, there must be an interest in both the scientific community and the pharmaceutical industry in investing in SEDDS for IN administration to make its commercialization possible in the near future.

## Figures and Tables

**Figure 1 pharmaceutics-14-01487-f001:**
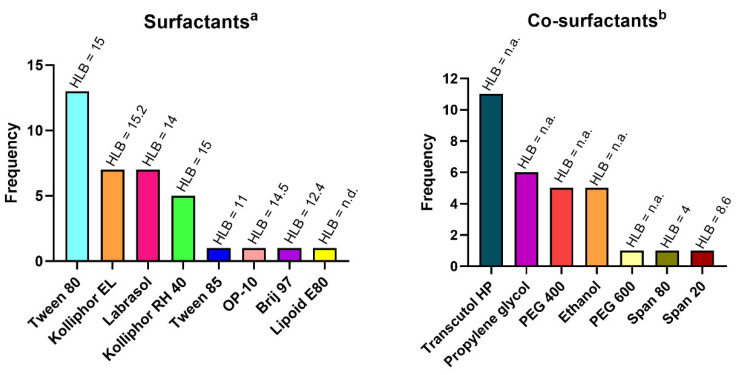
Frequency of surfactants (^a^ hydrophilic surfactants with HLB > 10) and cosurfactants (^b^ hydrophobic surfactants and organic solvents used other than oils with HLB < 10) used in the preparation of self-emulsifying drug delivery systems (SEDDS) described in the revised articles. Hydrophilic–lipophilic balance (HLB) is described for each excipient, except for those that are not determined (n.d.) or not appliable (n.a.). Created with GraphPad Prism software, version 8.0 (San Diego, CA, USA).

**Figure 2 pharmaceutics-14-01487-f002:**
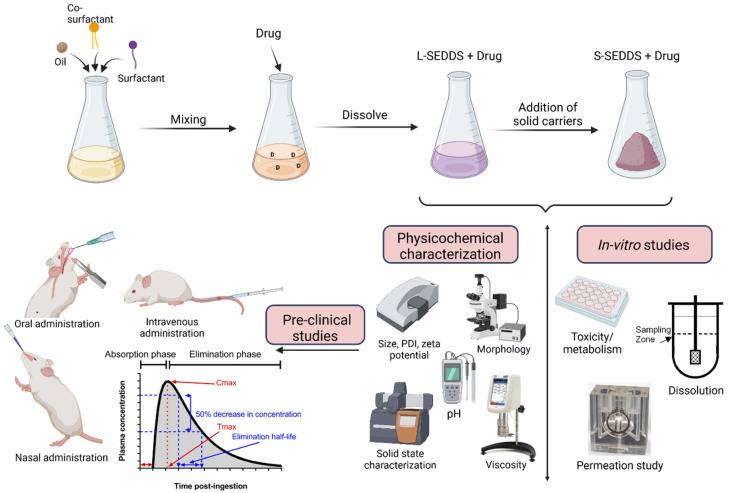
Preparation of self-emulsifying drug delivery systems (SEDDS) in the liquid (L-SEDDS) or solid (S-SEDDS) state together with the main techniques used for physicochemical characterization, in vitro evaluation, and implementation of in vivo studies. Created with BioRender.com, accessed on 17 June 2022.

**Figure 3 pharmaceutics-14-01487-f003:**
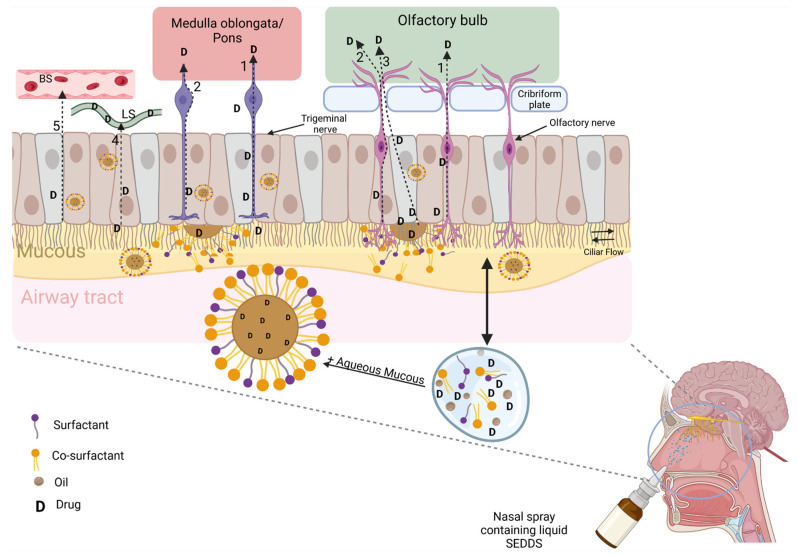
Intranasal delivery of drugs loaded in self-emulsifying drug delivery systems (SEDDS). After self-emulsification, drugs can be directly delivered to the brain through olfactory pathway—by intracellular (1), paracellular (2) or transcellular/intracellular (3) mechanisms—and through trigeminal pathways—by intracellular (1) or paracellular (2) mechanisms. Molecules can also reach the brain by indirect systemic pathways—from lamina propria, drugs can be absorbed by lymphatic system (LS) (4) or blood system (BS) (5)—reaching systemic circulation, crossing the blood–brain barrier and, in the end, the brain. Created with BioRender.com, accessed on 17 June 2022.

**Table 1 pharmaceutics-14-01487-t001:** Composition and main physicochemical characteristics of self-emulsifying drug delivery systems (SEDDS) investigated for brain delivery of neurotherapeutics by intranasal (IN), intravenous (IV) or oral routes. The methodology used for data collection is summarized in Appendix A of the Appendix A.

Drug	Administration Route	SEDDS Type ^1^	Components			Droplet Size (nm)	PDI	Zeta Potential (mV)	Viscosity (cP)	pH	Refs.
			Oil	Surfactant ^2^	Cosurfactant ^3^						
**Perampanel**	IN	SMEDDS	Miglyol 812	Kolliphor RH 40	Transcutol HP	20.07 ± 0.03	0.060 ± 0.001	NR	110.50 ± 1.05	5.9 ± 0.22	[28]
**α-pinene**	Oral	SNEDDS	Anise oil	Tween 80	Transcutol HP	11.79 ± 0.05 (1:50)	0.074 ± 0.003 (1:50)	16.37 ± 0.39 (1:50)	338.3 ± 0.001 (undiluted)	NR	[29]
						12.03 ± 0.08 (1:100)	0.093 ± 0.022 (1:100)	24.53 ± 0.17 (1:100)			
						11.69 ± 0.16 (1:200)	0.161 ± 0.019 (1:200)	30.6 ± 0.15 (1:200)	1.21 ± 0.004 (1:50)		
						11.64 ± 0.26 (1:400)	0.224 ± 0.019 (1:400)	33.20 ± 0.36 (1:400)	1.02 ± 0.005 (1:100)		
						11.61 ± 0.20 (1:800)	0.391 ± 0.029 (1:800)	34.63 ± 0.49 (1:800)			
**Huperzine A**	IN	SMEDDS	Castor oil	Kolliphor RH 40	Propylene glycol	21.26	0.234	−28.3	560 ± 10 at ambient conditions ^4^	5.5 ± 0.5 ^4^	[30]
						20.53 ^4^	0.168 ^4^	−21.9 ^4^	5200 ± 100 at physiologic conditions ^4^		
**Bdph**	IN	SNEDDS	Bdph ^5^	Kolliphor EL	PEG 400	2643.72 ± 1325.18	0.42 ± 0.20	−3.43 ± 0.20	66.9 ± 3.24	NR	[4]
**Ferulic acid**	Oral	SMEDDS	Glyceryl triacetate	OP-10 + Labrasol	PEG 400	15.79 ± 0.60	0.236 ± 0.039	NR	NR	NR	[31]
**Oxyresveratrol**	Oral	SMEDDS	Capryol 90	Kolliphor RH 40 + Tween 80	---	26.6 ± 0.1 (1:20 in water)	0.07 ± 0.01 (1:20 in water)	NR	NR	NR	[32]
						31.4 ± 0.2 (1:20 in gastric fluid pH 1.2)	0.08 ± 0.01 (1:20 in gastric fluid pH 1.2)				
						32.2 ± 0.1 (1:20 in apical medium pH 6.5)	0.09 ± 0.01 (1:20 in apical medium pH 6.5)				
						32.4 ± 0.2 (1:20 in basolateral medium pH 7.4)	0.08 ± 0.01 (1:20 in basolateral medium pH 7.4)				
**DPA**	Oral	SEDDS	Maisine 35-1 + Soya bean oil	Kolliphor RH 40	Ethanol	NR	NR	NR	NR	NR	[33]
**DHA**	Oral	SNEDDS	Olive oil	Tween 80	Propylene glycol+ Span 80 + Span 20	17.6 ± 3.5	0.202 ± 0.043	−37.6 ± 0.5	28.34 ± 1.5	7.1 ± 0.1	[25]
**Teniposide ^6^**	IV	SMEDDS	MCT	Lipoid E80	Ethanol	282 ± 21	0.423 ± 0.035	−7.5 ± 1.7	NR	NR	[34]
**Naringin**	IN	SNEDDS	Acrysol K140	Tween 80	Transcutol HP	152.03 ± 4.6	0.23	−15.0	NR	6.5 ± 0.2	[5]
**L- THP**	Oral	SMEDDS	Capryol 90	Kolliphor RH 40	Transcutol HP	NR	NR	NR	NR	NR	[35]
**Lamotrigine**	Oral	S-SNEDDS	Rose oil	Kolliphor EL + Tween 80	---	16.3 ± 0.15 (SNEDDS)	0.25 ± 0.018 (SNEDDS)	−7.97 (SNEDDS)	NR	NR	[23]
**Stiripentol**	Oral	SNEDDS	Ethyl oleate	Kolliphor RH 40	Propylene glycol	45.52 ± 1.99	0.076 ± 0.011	−21.67 ± 0.24	NR	NR	[36]
**Zaleplon**	Oral	SNEDDS	Labrafil	Labrasol	Transcutol HP	56.42 ± 0.64	NR	NR	NR	NR	[37]
**Cannabidiol**	Oral	SEDDS	^7^	^7^	^7^	40–50	<0.1	NR	NR	NR	[38]
**Sertraline HCl**	Oral	S-SNEDDS	Capmul MCM	Kolliphor EL	Transcutol HP	20.10 ± 1.93 (SNEDDS)	0.26 (SNEDDS)	−11.26	NR	NR	[39]
						168.00 ± 6.71 (S-SNEDDS)	0.47 (S-SNEDDS)	−17.00			
**Quercetin**	Oral	SNEDDS	Oleic acid	Tween 80	Transcutol HP + PEG 400	94.63 ± 3.17	NR	−17.91 ± 1.02	NR	NR	[40]
**Ziprasidone**	Oral	Pellet SNEDDS	Capmul MCM	Labrasol	PEG 400	[54.5–62.3] ^8^	NR	−28	NR	NR	[41]
**Olanzapine**	Oral	SNEDDS	Capryol 90	Brij 97	Ethanol	90	0.287	−19.0	22.3	NR	[42]
**Carbamazepine**	Oral	SS-SMEDDS ^9^	Miglyol 812N	Kolliphor EL + Tween 80	PEG 400	33.7	NR	NR	NR	NR	[43]
**Diazepam**	IN	SMEDDS	Ethyl laurate	Labrasol	Transcutol HP + Ethanol	48.1 ± 4.5	NR	NR	NR	NR	[44]
**Clonazepam**	IN	SMEDDS	MCT	Kolliphor EL + Tween 80	Propylene glycol	15.21	NR	−29.88	NR	NR	[45]
**Zolmitriptan**	Oral	SNEDDS	Lavender	Kolliphor EL	Transcutol HP	19.59 ± 0.36	0.29 ± 0.009	−23.5 ± 1.17	NR	NR	[46]
**Ginsenoside Rg1**	Oral	SNEDDS (F1)	Isopropyl myristate	Tween 80	Transcutol HP	10.05 ± 2.3	0.119	−9.92 ± 5.2	54.5 ± 4.5	NR	[47]
		SNEDDS (F11)	Capryol 90	Tween 80	Transcutol HP	10.90 ± 2.1	0.121	−7.38 ± 4.5	53.7 ± 2.1	NR	
**Chlorogenic acid ^10^**	Oral	SMEDDS	Ethyl oleate	Labrasol	Transcutol HP	66.5 ± 1.3	~0.2	−8.4 ± 0.6	NR	NR	[48]
**CAT3**	Oral	SMEDDS	Isopropyl Myristate	Kolliphor EL + Labrasol	---	26.93 ± 0.22 (1 mg/mL CAT3)	<0.3	Negative for blank SMEDDS	NR	NR	[49]
						14.94 ± 0.05 (10 mg/mL CAT3)		Positive for CAT3-SMEDDS			
**Curcumin**	Oral	SMEDDS	Oleic acid	Tween 80	Propylene glycol	44.13 ± 0.695	0.446	−25.43 ± 0.94	NR	NR	[50]
**Chlorpromazine**	Oral	SNEDDS	Olive Oil:Linseed Oil (1:2, *w/w*)	Tween 85	Ethanol	178 ± 16	0.31 ± 017	−21.4	NR	7.4 ± 1.0	[6]
**Puerarin + Borneol**	Oral	SMEEDS	Capmul MCM	Tween 80 + Labrasol	Propylene glycol	151.6 ± 1.92	NR	−4.73 ± 0.38	NR	NR	[51]
**Levosulpiride**	Oral	SNEDDS	Castor oil	Tween 80	PEG 600	197.3	0.301	−18.8	12	NR	[24]

Bdph, Butyldenephthalide; CAT3, 13a-(S)-3-pivaloylocyl-6,7-dimethoxyphenanthro(9,10-b)-indolizidine; DHA, docosahexaenoic acid; DPA, dipalmitoyl-apomorphine; L-THP, L-Tetra hydropalmatine; MCT, medium-chain triglyceride; NR, not reported; PEG 400, polyethylene glycol 400; SEDDS, self-emulsifying drug delivery system; SMEDDS, self-microemulsifying drug delivery system; SNEDDS, self-nanoemulsifying drug delivery system; SS-SMEDDS, supersaturable-SMEDDS. ^1^ Characterization of SEDDS type based on authors nomenclature; ^2^ hydrophilic surfactants with HLB > 10; ^3^ organic solvents and hydrophobic surfactants used other than oils with HLB < 10; ^4^ values referent to the SMEDDS temperature and pH-responsive in situ gel; ^5^ yellow oily liquid dissolved in 6.5% DMSO; ^6^ dissolved in N,N-dimethylacetamide before incorporation in SMEDDS; ^7^ SEDDS composition of VESIsorb^®^, a Swiss patented technology for improving the bioavailability of poorly absorbed ingredients; ^8^ interval range of droplet sizes obtained after different dilution ratios in different dilution mediums; ^9^ contains 2% *w*/*w* of PVP K90 as a precipitation inhibitor; ^10^ chlorogenic acid–phospholipid complex with soybean phospholipid to associate chlorogenic acid with oil droplets of SMEDDS, improving lipophilicity and intestinal permeation.

**Table 2 pharmaceutics-14-01487-t002:** Pre-clinical studies for evaluation of in vivo outcomes after administration of neurotherapeutics loaded in self-emulsifying drug delivery systems (SEDDS). The methodology used for data collection is summarized in Appendix A of the Appendix A.

Drug	BCS Class	Therapeutic Use	Administration Route	Limitations	In Vivo Study	In Vivo Main Outcomes	Refs.
**Perampanel**	NR	Epilepsy	IN	Decrease dose; increase patient chronic compliance; allow the use in status epilepticus.	ICR mice	Comparison with oral suspension: shorter t_max_; plasmatic and brain AUC_0-t_ 1.4- and 1.6-fold higher; plasmatic and brain C_max_ 2.3- and 3.3-fold higher; F_rel_ of 134.1%; DTE = 116.3% and DTP = 14.3%. No histopathological toxicity in nasal mucosa after a 7-day repeated dose.	[28]
**α-pinene**	NR	Neuroprotective (Parkinson’s disease)	Oral	Low absorption; fast metabolism and elimination.	Wistar rats	Improved physical and behavior activities; acute toxicity only with 2000 mg/kg.	[29]
**Huperzine A**	NR	Alzheimer’s disease	IN	Low aqueous solubility and bioavailability; fluctuations in blood concentration and related side effects.	Sprague Dawley rats	Comparison with IV solution: plasmatic and brain absolute bioavailability of 122.55% and 120.38%; sustained released behavior shown by higher t_1/2el_ and MRT values.	[30]
**Bdph**	NR	Glioblastoma	IN	Low aqueous solubility and spreadability; first-pass metabolism.	Fisher 344 rats	Half the dose required for the same therapeutic effect (SNEDDS (160 mg/kg), solution (320 mg/kg)).	[4]
**Ferulic acid**	NR	Insomnia	Oral	Low stability in GIT and bioavailability; short t_1/2_.	Wistar rats	Plasmatic C_max_ and AUC_0–8 h_ 1.1- and 1.7-fold higher; increase in t_1/2_ (2.1 h vs. 1.39 h); F_rel_ of 185.96%; decrease in kidney distribution from 76.1% to 59.4%.	[31]
					Kunming mice	High ferulic acid distribution and enhanced serotonin levels in the brain; extended sleep time by 2.0-fold in insomnia mice.	
**Oxyresveratrol**	NR	Antioxidant/Neuroprotective (Alzheimer’s disease)	Oral	Poor intestinal permeability and very low bioavailability; P-gp efflux; first-pass metabolism; fast elimination.	Wistar rats	Plasmatic C_max_ and AUC_0–10 h_ 3.6- and 7.9-fold higher with SMEDDS than with suspension. 4.0-fold decrease in the dose required for neurotoxicity prevention.	[32]
					ICR mice		
**DPA**	NR	Parkinson’s disease	Oral	Short plasma t_1/2_; lack of compliance to subcutaneous injections; extensive first-pass effect.	Sprague Dawley rats	Response duration in lesioned animals increased to 6 h with SEDDS (2.5 h with oral apomorphine, oil-in-water DPA emulsion and 1 h with subcutaneous apomorphine).	[33]
**DHA**	NR	Neurodevelopment (in pregnancy and early childhood)	Oral	Soft gelatin capsules are not fit for children; poor dispersibility, solubility, organoleptic properties, and compliance.	Albino rats	Brain concentration increased 2.6-fold comparatively with the marketed formulation; enhanced performance activity in rats treated with SNEDDS.	[25]
**Teniposide**	NR	Neuroblastoma/Cerebroma	IV	Poor aqueous solubility; instability in aqueous solution; systemic toxicity caused by commercial injection.	Sprague Dawley rats	Comparison with commercial injection: plasmatic C_max_ and AUC_0-__inf_ significantly lower; clearance and distribution volume significantly higher; high brain teniposide accumulation.	[34]
**Naringin**	II	Neuroprotective (Alzheimer’s and Parkinson’s diseases)	IN	Low aqueous solubility; poor bioavailability.	Wistar rats	Comparison with nasal suspension: 2.6- and 7.1-fold increase in blood and brain AUC_0–6 h_; 2.4- and 3.0-fold increase in blood and brain C_max_. DTE = 566.11%; DTP = 82.3%; F_rel_ of 306.6%. No histopathological toxicity in nasal sheep mucosa after 6 h of treatment.	[5]
**L-THP**	II	Cocaine addiction	Oral	Poor aqueous solubility and membrane absorption; high pharmacokinetic variability.	Sprague Dawley rats	Comparison with oral suspension: 0.7-fold lower brain C_max_; 3.5-fold higher brain AUC_0–24 h;_ 3.3-fold increase in F_rel_.	[35]
**Lamotrigine**	II	Epilepsy	Oral	Low aqueous solubility and oral bioavailability; precipitation in the small intestine.	New Zealand rabbits	Comparison with pure drug and Lamictal^®^: F_rel_ of 203.31% and 160.53%; plasmatic C_max_ and AUC_0-inf_ 2.0- and 1.5-fold higher with S-SNEDDS.	[23]
**Stiripentol**	II	Epilepsy/Dravet syndrome	Oral	Poor aqueous solubility; gastric instability; slow and incomplete GIT dissolution.	Sprague Dawley rats	Comparison with suspension: plasmatic C_max_ and AUC_0–6 h_ 2.1- and 2.2-fold higher; brain C_max_ 2.32-fold higher; F_rel_ of 218.01%.	[36]
**Zaleplon**	II	Insomnia	Oral	Poor aqueous solubility and dissolution rate; delayed onset of action; first-pass metabolism.	Humans	Comparison with commercial product: plasmatic C_max_ and AUC_0–4 h_ 1.3-fold higher; decrease in t_max_ (0.506 h vs. 1.027 h).	[37]
**Cannabidiol**	II	Epilepsy/Depression/Anxiety/Psychosis/Analgesia/Neuroprotection	Oral	Poor aqueous solubility; extensive first-pass metabolism; food effect in absorption; erratic bioavailability.	Humans	Comparison with reference formulation: plasmatic AUC_0–8 h_ and AUC_0–24 h_ 2.9- and 1.7-fold higher; 4.4-fold increase in plasmatic C_max_; faster absorption with SEDDS (t_max_ of 1 h vs. 3 h).	[38]
**Sertraline HCl**	II	Depression/Anxiety	Oral	Poor aqueous solubility and bioavailability; extensive first-pass metabolism; GIT side effects.	Humans	Comparison with commercial tablets: AUC_0–72 h_ 1.6-fold higher; high MRT values for S-SNEDDS (28.16 ± 0.82 h vs. 24.24 ± 1.58 h).	[39]
**Quercetin**	II	Cerebral ischemia	Oral	Poor aqueous solubility, permeability, and bioavailability; highly degradable in GIT; extensive first-pass effect.	Wistar rats	Striatum C_max_ 9.4-fold higher with SNEDDS than with oral solution.	[40]
**Ziprasidone**	II	Schizophrenia and bipolar disorder	Oral	Low bioavailability; highly susceptible to food effect causing fluctuations in plasmatic concentrations.	Dogs	Plasmatic t_max_ increased to 6.1 ± 1.0 h and 5.8± 1.1 h in the fed and fasted states; MRT increase to 11.0 ± 5.5 h in the fed state; no food effect was obtained with pellet SNEDDS oral administration; F_rel_ of 157.8% and 150.1% in fed and fasted states.	[41]
**Olanzapine**	II	Antipsychotic	Oral	Poor aqueous solubility; first-pass metabolism; high doses required.	New Zealand rabbits	Comparison with suspension and tablets: plasmatic C_max_ 1.4- and 1.3-fold higher; plasmatic AUC_0–12 h_ 1.5- and 1.3-fold higher; faster t_max_ (1 h vs. 2 h).	[42]
**Carbamazepine**	II	Epilepsy	Oral	Poor aqueous solubility; slow and irregular GIT absorption; high variability in plasma concentrations.	Beagle dogs	Plasmatic C_max_ and AUC_0–12 h_ 6.7- and 5.9-fold higher with SS-SMEDDS than with tablets.	[43]
**Diazepam**	II	Epilepsy	IN	Parental and rectal routes for diazepam administration in status epilepticus.	New Zealand rabbits	T_max_ of 10 min with IN SMEDDS; bioavailability of 51.6% and 45.9% in rabbit plasma and brain; AUC_brain_/AUC_plasma_ ratios lower after IN than IV administration (3.77 ± 0.17 vs. 4.23 ± 0.08).	[44]
					Sprague Dawley rats	T_max_ of 5 min with IN SMEDDS; bioavailability of 68.4% and 67.7% in rat plasma and brain; similar AUC_brain_/AUC_plasma_ ratios after IN and IV administrations; molecules possibly reach the brain by systemic route after IN administration.	
**Clonazepam**	II	Epilepsy	IN	Only IV route available for status epilepticus; limited uptake to the brain after oral administration.	Swiss albino rats	Brain C_max_ and AUC_0-inf_ 2.2- and 1.2-fold higher with SMEDDS than with IN solution; DTE = 131% and DTP = 20%.	[45]
**Zolmitriptan**	III	Migraine	Oral	Low permeability; hepatic first-pass effect; low oral bioavailability; severe adverse effects.	Wistar rats	Nontoxic effects after 14 days of treatment; lower brain concentration with SNEDDS maintaining the same effect in physiological state, algesia.	[46]
**Ginsenoside Rg1**	III	Obesity (by neurotransmission regulation)	Oral	Low membrane permeability.	Albino rats	Shorter t_max_; brain C_max_ and AUC_0–12 h_ 2.6- and 3.3-fold higher than suspension; higher weight loss.	[47]
**Chlorogenic acid**	III	Glioblastoma (by immunomodulation activity)	Oral	Low membrane permeability; widely metabolized by gut flora; daily intramuscular injection to attain lymph nodes.	Sprague Dawley rats	Plasmatic AUC_0–8 h_ and C_max_ 5.1- and 9.5-fold higher with SMEDDS than with suspension.	[48]
					Beagle dogs	Oral absolute bioavailability 2.5-fold higher with SMEEDS than with suspension.	
					ICR mice	Inhibition of tumor growth in glioma model through immunomodulation.	
**CAT3**	IV	Glioblastoma	Oral	Insoluble in water; low bioavailability; metabolized in intestinal fluid with; severe GIT side effects.	Sprague Dawley rats	AUC_0–24 h_, C_max_ and MRT in plasma 1.8-, 0.40-, and 1.7-fold higher with SMEDDS than with suspension.	[49]
					ICR mice	Stronger antiglioma effect with no black coloration and necrosis in mice GIT receiving SMEDDS.	
**Curcumin**	IV	Depression	Oral	Low aqueous solubility, intestinal permeability and bioavailability; fast hepatic metabolism.	Wistar rats	Open field test: significant increase in frequency and duration spent in the central area;passive avoidance test: decrease in step down avoidance; ambulation counts test: animal movements increased.	[50]
**Chlorpromazine**	IV	Antipsychotic/antiemetic	Oral	Low aqueous solubility, permeability, and bioavailability; extensive first-pass metabolism.	Sprague Dawley rats	Comparison with suspension: plasmatic C_max_ and AUC_0–24 h_ 3.7- and 6.0-fold higher; t_1/2_ significantly higher with SNEDDS (9.88 ± 0.25 h vs. 5.74 ± 0.31 h); oral bioavailability 6.5-fold higher.	[6]
**Puerarin + Borneol**	IV	Cerebral ischemia	Oral	Puerarin: poor aqueous solubility and permeability; P-gp substrate; poor BBB penetration; acute side effects with IV injection.	Kumming mice	Comparison with NCS and ICS: plasmatic C_max_ 1.35- and 2.34-fold higher; AUC_0–12 h_ 1.7- and 2.3-fold higher; brain C_max_ 2.1- and 1.2-higher; AUC_0–12 h_ 1.7- and 1.5-fold higher; brain t_1/2el_ increased 159.54%.	[51]
**Levosulpiride**	IV	Schizophrenia	Oral	Low aqueous solubility and oral bioavailability.	Albino rats	Comparison with tablets: plasmatic C_max_ and AUC_0–36 h_ 1.7- and 1.5-fold higher; decrease in t_max_ (3 h vs. 4 h).	[24]

AUC_0-t_, area under the concentration–time curve from time zero to the time of the last quantifiable drug concentration; AUC_inf_, area under drug concentration–time curve from time zero to infinity; BBB, blood–brain barrier; Bdph, butylidenephthalide; CAT3, 13a-(S)-3-pivaloylocyl-6,7-dimethoxyphenanthro(9,10-b)-indolizidine; C_max_, maximum (peak) concentration; DHA, docosahexaenoic acid; DPA, dipalmitoyl-apomorphine; DTE, drug targeting efficiency; DTP, direct transport percentage; F_rel_, relative bioavailability; GIT, gastrointestinal tract; ICS, inclusion compounds solution; IN, intranasal; IV, intravenous; LTHP, L-Tetra hydropalmatine; MRT, mean residence time; NCS, nanocrystals suspension; P-gp, P-glycoprotein; SEDDS, self-emulsifying drug delivery system; SMEDDS, self-microemulsifying drug delivery system; SNEDDS, self-nanoemulsifying drug delivery system; S-SNEDDS, solid self-nanoemulsifying drug delivery system; SS-SMEDDS, supersaturable-SMEDDS t_1/2el_, elimination half-life; t_max_, time to reach maximum (peak) concentration.

## Data Availability

Not applicable.

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
