# Peer review of "Self-Emulsifying Drug Delivery Systems: An Alternative Approach to Improve Brain Bioavailability of Poorly Water-Soluble Drugs through Intranasal Administration"

_pharmaceutics, 2022, doi:10.3390/pharmaceutics14071487_

Round 1
Reviewer 1 Report
The manuscript describes the potential application of self-emulsifying drug delivery systems (SEDDS) for the nose-brain or oral delivery of neurotherapeutic drugs. The manuscript is very interesting and well write; it will help the researcher to use the correct terminology to describe the component of these formulations. So, I recommend the publication in Pharmaceutics after minor revision.
In particular:
- The manuscript reports the similarities and differences between SEDDS intended for nasal or oral administration; for that, “oral” could be added in the title.
- In Table 1, Authors should report for which administration route (nasal or oral) the formulation has been studied. Furthermore, a logical order should be given for the inclusion of drugs in the table: for example, first BCS II drugs and then BCS IV drugs.
- Line 182: Concerning the emulsifying agents, Authors should be reported their safety on the nasal epithelium as well as of co-surfactant.
- Please change Figure 3 to take into account that “trigeminal neuron dendrites do not completely penetrate the nasal epithelium in the same manner as olfactory neurons. Instead, they merely extend into the paracellular space, terminating shy of the apical surface of the epithelium (read Crowe et al https://doi.org/10.1016/j.lfs.2017.12.025 ).” On the contrary, the olfactory neurones project cilia into the mucous layer.
- Please modify Table 2 considering the same logical point of view of Table 1.
Author Response
Reviewer 1 comments:
The manuscript describes the potential application of self-emulsifying drug delivery systems (SEDDS) for the nose-brain or oral delivery of neurotherapeutic drugs. The manuscript is very interesting and well write; it will help the researcher to use the correct terminology to describe the component of these formulations. So, I recommend the publication in Pharmaceutics after minor revision.
Authors’ response: Thank you very much for your review and we are glad to know that you recommend the publication of our manuscript in Pharmaceutics. Please, find the response to your particular comments below.
- The manuscript reports the similarities and differences between SEDDS intended for nasal or oral administration; for that, “oral” could be added in the title.
Authors’ response: We gladly appreciate your suggestion. However, we decided not include “oral” in the manuscript title because we want to give a special focus to intranasal, and not oral route. The topic of intranasal administration of SEEDS is really new, in contrast to SEDDS aimed to be orally administrated. However, as the number of intranasal SEDDS reported in literature is still scarce, we intended to use the oral route as a reference to clarify the interest of using SEDDS for neurologic diseases treatment. That is why we did not discuss the mechanisms of emulsions formation and drug transport after SEDDS oral administration, as well as we did not deeply discuss all the referenced studies that developed an oral SEDDS, contrary to what occurred with all the intranasal SEDDS presented in this review.
- In Table 1, Authors should report for which administration route (nasal or oral) the formulation has been studied. Furthermore, a logical order should be given for the inclusion of drugs in the table: for example, first BCS II drugs and then BCS IV drugs.
Authors’ response: Thank you very much for your recommendation. In line with it, we add a column in Table 1 to clarify the intended route of administration of each presented SEDDS. As you also suggested, we change the order of drugs inclusion in both Table 1 and Table 2, now appearing as BCS class not reported (NR) > class II > class III > class IV.
- Line 182: Concerning the emulsifying agents, Authors should be reported their safety on the nasal epithelium as well as of co-surfactant.
Authors’ response: Thank you so much for your commentary. In fact, safety concerns regarding SEDDS excipients are important and is of the utmost importance to evaluate possible toxic effects to avoid adverse effects after intranasal administration. To highlight it and to make researchers aware of this importance, we add some important remarks regarding this issue. Please see line 180-182 of the new manuscript version.
- Please change figure 3 to take into account that “trigeminal neuron dendrites do not completely penetrate the nasal epithelium in the same manner as olfactory neurons. Instead, they merely extend into the paracellular space, terminating shy of the apical surface of the epithelium (read Crowe et al https://doi.org/10.1016/j.lfs.2017.12.025 ).” On the contrary, the olfactory neurones project cilia into the mucous layer.
Authors’ response: We are grateful for your comment. Following the information provided by the Crowe et. al., we change Figure 3 in accordance to what you referred in the commentary and that is also described in the article. Please find out the new Figure 3 in the latest version of the manuscript.
- Please modify Table 2 considering the same logical point of view of Table 1.
Authors’ response: Once again, thank you for your recommendation. As already referred in the answer to your second comment, we changed it in accordance with you suggestion.
Reviewer 2 Report
1. The Title of the paper should modified to poorly water- soluble drugs instead of poor water soluble drugs
also remove (in) from the title
2. The English language and style should be revised
3. The number of digits in the tables should be unified
4. Nasal tolerability studies should mentioned in the review article
Author Response
Reviewer 2 comments:
- The Title of the paper should modified to poorly water- soluble drugs instead of poor water soluble drugs. Also remove (in) from the title.
Authors’ response: Thank you very much for your suggestion and alert. The Title was modified according to the requested.
- The English language and style should be revised
Authors’ response: We appreciate your commentary and, accordingly, we revised all English language and style throughout the manuscript.
- The number of digits in the tables should be unified
Authors’ response: Thank you for your suggestion. We revised all digits in the tables and unified the values calculated by us (e.g. “brain AUC0-t 1.4- and 1.6-fold higher”). However, we decided not to change the remaining values as they correspond to the values reported in the original reviewed articles (e.g. “bioavailability of 122.55% and 120.38%”).
- Nasal tolerability studies should mentioned in the review article.
Authors’ response: Thank you very much for your commentary and suggestion. In fact, tolerability assessment of SEDDS is of utmost importance regarding possible local side effects of these formulations, particularly in nasal mucosa. Consequently, we now highlight that importance in the last paragraph of section 2.2. of the new version of the manuscript. Moreover, we also revised once more all the studies that developed an intranasal SEDDS, including the nasal tolerability findings of those studies either in Table 2 and along the text in section 4.
Reviewer 3 Report
In the review, the authors collected good evidence about the benefits of the drug delivery system indicated in the title. In addition, a lot of work has been done to systematize information about the physicochemical parameters of dosage forms, their effect on the body of laboratory animals. However, it would be nice to include in the review the methodology by which the sources were searched, in which databases, what is the total number of articles on this topic presented. There are drawings in the manuscript, but there is no indication of the original source in the caption to them. Perhaps they are original. Please explain.
Author Response
Reviewer 3 comments
In the review, the authors collected good evidence about the benefits of the drug delivery system indicated in the title. In addition, a lot of work has been done to systematize information about the physicochemical parameters of dosage forms, their effect on the body of laboratory animals. However, it would be nice to include in the review the methodology by which the sources were searched, in which databases, what is the total number of articles on this topic presented. There are drawings in the manuscript, but there is no indication of the original source in the caption to them. Perhaps they are original. Please explain.
Authors’ response: Thank you so much for your commentaries. Following your suggestion, we created a diagram – please see Figure S1 of the Supplementary material - in which are exposed the databases used, the search terms applied and the number of articles found that developed SEDDS for brain targeting of neurotherapeutics. We also add a reference to this new figure in the captions of Table 1 and 2 (please see new version of the manuscript). Regarding your question about the figures present in the manuscript, all of them are original. We create Figure 1 in GraphPad v8.0 and the remaining were created in Biorender.com. To clarify it, we add that information to each figure caption.
Round 2
Reviewer 2 Report
Accept the review article